# A Phenylfurocoumarin Derivative Reverses ABCG2-Mediated Multidrug Resistance In Vitro and In Vivo

**DOI:** 10.3390/ijms222212502

**Published:** 2021-11-19

**Authors:** Shoji Kokubo, Shinobu Ohnuma, Megumi Murakami, Haruhisa Kikuchi, Shota Funayama, Hideyuki Suzuki, Taiki Kajiwara, Akihiro Yamamura, Hideaki Karasawa, Norihiko Sugisawa, Kosuke Ohsawa, Kuniyuki Kano, Junken Aoki, Takayuki Doi, Takeshi Naitoh, Suresh V. Ambudkar, Michiaki Unno

**Affiliations:** 1Department of Surgery, Tohoku University Graduate School of Medicine, Sendai 980-8574, Japan; shojikokubo2016@surg.med.tohoku.ac.jp (S.K.); megu.591118@gmail.com (M.M.); hideji@surg.med.tohoku.ac.jp (H.S.); t-kajiwara@surg.med.tohoku.ac.jp (T.K.); akihiro-y@surg.med.tohoku.ac.jp (A.Y.); h-karasawa@surg.med.tohoku.ac.jp (H.K.); n.sugisawa@surg.med.tohoku.ac.jp (N.S.); naitot@med.kitasato-u.ac.jp (T.N.); m_unno@surg.med.tohoku.ac.jp (M.U.); 2Laboratory of Cell Biology, Center for Cancer Research, National Cancer Institute, NIH, Bethesda, MD 20892, USA; ambudkar@mail.nih.gov; 3Graduate School of Pharmaceutical Sciences, Tohoku University, Sendai 980-8578, Japan; halkiku@keio.jp (H.K.); shota.funayama.t1@dc.tohoku.ac.jp (S.F.); kosuke@mail.pharm.tohoku.ac.jp (K.O.); k-kano@mol.f.u-tokyo.ac.jp (K.K.); jaoki@mol.f.u-tokyo.ac.jp (J.A.); oi_taka@mail.pharm.tohoku.ac.jp (T.D.)

**Keywords:** ABCG2 inhibitor, multidrug resistance, phenylfurocoumarin, ABC transporter, chemosensitivity

## Abstract

The ATP-binding cassette subfamily G member 2 (ABCG2) transporter is involved in the development of multidrug resistance in cancer patients. Many inhibitors of ABCG2 have been reported to enhance the chemosensitivity of cancer cells. However, none of these inhibitors are being used clinically. The aim of this study was to identify novel ABCG2 inhibitors by high-throughput screening of a chemical library. Among the 5812 compounds in the library, 23 compounds were selected in the first screening, using a fluorescent plate reader-based pheophorbide a (PhA) efflux assay. Thereafter, to validate these compounds, a flow cytometry-based PhA efflux assay was performed and 16 compounds were identified as potential inhibitors. A cytotoxic assay was then performed to assess the effect these 16 compounds had on ABCG2-mediated chemosensitivity. We found that the phenylfurocoumarin derivative (R)-9-(3,4-dimethoxyphenyl)-4-((3,3-dimethyloxiran-2-yl)methoxy)-7H-furo [3,2-g]chromen-7-one (PFC) significantly decreased the IC_50_ of SN-38 in HCT-116/BCRP colon cancer cells. In addition, PFC stimulated ABCG2-mediated ATP hydrolysis, suggesting that this compound interacts with the substrate-binding site of ABCG2. Furthermore, PFC reversed the resistance to irinotecan without causing toxicity in the ABCG2-overexpressing HCT-116/BCRP cell xenograft mouse model. In conclusion, PFC is a novel inhibitor of ABCG2 and has promise as a therapeutic to overcome ABCG2-mediated MDR, to improve the efficiency of cancer chemotherapy.

## 1. Introduction

Multidrug resistance (MDR) remains a major problem in anticancer therapy. It often occurs when cancer patients acquire resistance to many functionally and structurally unrelated anticancer drugs [1]. There are several different mechanisms involved in MDR, including the overexpression of ATP-binding cassette (ABC) transporters [2,3]. ABC transporters transport molecules via an energy-dependent process involving the hydrolysis of ATP. Currently, the 48 ABC transporters have been classified into seven subfamilies (ABCA-ABCG), based on their structures [4]. Among them, ABCB1 (P-glycoprotein; P-gp, MDR1), ABCC1 (multidrug resistance-associated protein 1; MRP1), and ABCG2 (breast cancer resistance protein; BCRP, mitoxantrone resistance protein; MXR) are the major transporters involved in MDR, in cancer cells.

ABCG2 appears to protect the body from toxicity associated with xenobiotic exposure. ABCG2 was originally discovered in human placenta, drug-resistant breast cancer cells, and drug-resistant colon cancer cells [5,6,7]. ABCG2 is normally expressed in many tissues, including the placenta, blood–brain barrier, blood–testis barrier, liver, kidney, gastrointestinal tract, and mammary glands [8,9,10,11].

However, ABCG2 is overexpressed in cancer cells and plays a crucial role in the development of MDR. Many ABCG2 inhibitors have been reported to date, including mitoxantrone [6], flavopiridol [12], as well as the camptothecins topotecan, irinotecan, and its active metabolite, SN-38 [13]. In addition, tyrosine kinase inhibitors, including imatinib and gefitinib, have also been reported as substrates of ABCG2 [14,15].

Fumitremorgin C, isolated from *Aspergillus fumigatus*, was identified as the first inhibitor of ABCG2 [16]; however, its neurotoxicity precluded its clinical use and several other analogs were developed. Among these analogs, Ko143 was found to be a more potent selective inhibitor of ABCG2 [17]. Although fumitremorgin C and Ko143 are selective inhibitors of ABCG2, there are other non-selective inhibitors of ABCG2, including elacridar [18], tariquidar [19], cyclosporin A [20], and curcumin [21], which inhibit not only ABCG2, but also ABCB1. Over several years, some studies have reported novel ABCG2 inhibitors [22,23,24,25,26,27], but there have not been clinical trials demonstrating the reversal of ABCG2-mediated MDR due to toxicity and instability in vivo [28].

To identify novel ABCG2 inhibitors from a different perspective, some groups have performed high-throughput screening using a chemical library [29,30,31,32]. Based on these reports, a structure-based study and a high-throughput assay-based experiment seemed to be effective to identify novel inhibitors. Since Tohoku University has a chemical library that yielded a novel potent ABCB1 modulator in a previous study [33], the same chemical library was used in this study.

The aim of this study was to identify novel ABCG2 inhibitors in our chemical library. Subsequent validation assays, functional analyses, and in vivo assays have led to the discovery of new potent inhibitors of ABCG2 that has potential for clinical use.

## 2. Results

### 2.1. High-Throughput Screening

From the chemical library, 5812 compounds were assessed using a fluorescent plate reader-based PhA efflux assay. The results of the primary high-throughput screening are presented as a dot plot in Figure 1A. Twenty-three compounds exhibited higher fluorescence intensity than the positive control, Ko143 (60%) and were candidates selected for analysis.

### 2.2. Flow Cytometry Assay

To analyze candidate compounds, flow cytometry was performed. As there was an overlap in one of the compounds, the number of candidate compounds that we worked with was reduced to 22. At a concentration of 10 µM, six compounds did not inhibit ABCG2, as shown in Figure 1B (lower fluorescence intensity than 30% fluorescence intensity of the positive control, Ko143). Therefore, the 16 other compounds were selected for assessment in subsequent cytotoxic assays.

### 2.3. Cytotoxic Assay

To assess the effect of the 16 selected candidates on the sensitivity of ABCG2-expressing cells to SN-38 (an ABCG2 substrate and an active metabolite of irinotecan), a cytotoxic assay was performed using MTS. Among the 16 compounds, a phenylfurocoumarin derivative (PFC) (Figure 2A) significantly decreased the IC_50_ of SN-38 in HCT-116/BCRP cells in a dose-dependent manner (Figure 2B and Table 1). PFC (10 μM, IC_50_: 0.19 ± 0.018 µM, fold reversal: 13.8) enhanced SN-38-induced cytotoxicity more significantly than Ko143 (1 µM, IC_50_: 0.26 ± 0.027 µM, fold reversal: 10.1). Whereas, for HCT-116 cells, fold reversals of 10 µM PFC and 1 µM Ko143 were 1.9 and 1.0, respectively (Figure 2C and Table 1). The data for the other compounds are provided in the Appendix A. To assess the cytotoxicity of Ko143 and PFC, an MTS assay was performed (Table 2). In both HCT-116 and HCT-116/BCRP cells, the concentrations of these compounds used in vitro did not cause toxicity.

### 2.4. ATPase Assay

Substrate-stimulated ATP hydrolysis of ABCG2 by PFC was measured in the total membranes of the ABCG2-expressing High Five insect cells. As shown in Figure 3A, PFC stimulated ATPase activity of ABCG2 in a concentration-dependent manner and the concentration of PFC required for 50% stimulation (EC_50_), was 3.8 ± 0.64 nM.

### 2.5. Reverse Transcription-Quantitative Polymerase Chain Reaction

The mRNA expression levels of *ABCG2* in HCT-116, HCT-116/BCRP, and HCT-116/BCRP cells treated with PFC, were evaluated using RT-qPCR. To assess the effect of PFC, the expression of *ABCG2* in HCT-116/BCRP and HCT-116/BCRP cells with PFC was compared to that of HCT-116 cells. As shown in Figure 3B, there was no significant difference in expression of *ABCG2* between cells treated with PFC for 3 d and untreated control cells. PFC did not inhibit ABCG2 expression.

### 2.6. Antitumor Activity In Vivo

The efficacy of PFC in the reversal of resistance to irinotecan was evaluated in the ABCG2-overexpressing HCT-16/BCRP cell xenograft model. As shown in Figure 4A, no significant difference was observed in tumor growth between the saline and PFC groups. However, tumor growth between the irinotecan and combination treatment groups showed significant differences on days 12, 15, 18, and 21. In addition, neither significant body weight loss (Figure 4B) nor adverse effects were observed in any group.

## 3. Discussion

There is a close relationship between the overexpression of ABCG2 and the development of MDR in lung, colon, and breast cancer [34]. Previous studies have reported that multiple compounds inhibit the transport function of ABCG2. However, no ABCG2 inhibitors with clinical efficacy have been approved because of their toxicity or limited bioavailability. For example, fumitremorgin C is a potent and specific ABCG2 inhibitor in vitro, but it causes neurotoxicity in vivo [16,35,36], which prevents its clinical use. Ko143 is also a potent inhibitor, but its limited bioavailability would hinder clinical use [37]. Recently, other ABCG2 inhibitors have been reported to have no clinical effects [38,39].

In this study, we identified a PFC as a novel potent ABCG2 inhibitor from our own chemical library. This PFC, which was obtained from *Angelicae dahuricae,* has not been previously reported to inhibit ABCG2. *A. dahuricae* is a traditional Chinese medicine that has been widely used to treat colds and headaches [40]. It has been reported that active compounds derived from *A. dahuricae*, including oxypeucedanin, have anti-inflammatory, antimicrobial, and antitumor activities [41,42,43]. PFC is structurally similar to oxypeucedanin and they both have common furocoumarin structures. As the furocoumarin structure appears to be important for the inhibition of ABCG2, the chemosensitivity of oxypeucedanin was also assessed. To evaluate the effect of oxypeucedanin on ABCG2 function, a cytotoxic assay was performed using the same method as described above. Oxypeucedanin (fold reversal at 10 μM: 3.0, fold reversal at 5 µM: 2.38) slightly enhanced the SN-38-induced cytotoxicity; however, its effect was lower than that of PFC and Ko143 (Appendix A). This emphasizes that it is the phenylfurocoumarin structure, not the furocoumarin structure, that plays an important role in the interaction with ABCG2.

Cytotoxic assays revealed that PFC increased the sensitivity of HCT-116/BCRP ABCG2-expressing cells to SN-38 (Figure 2B, Table 1). In addition, PFC did not affect the IC_50_ of SN-38 in parent HCT-116 cells (Figure 2C, Table 1). This indicates that PFC enhances the cytotoxicity of SN-38 in ABCG2-expressing cells by interacting with ABCG2. Furthermore, PFC at 5 or 10 µM did not cause any significant cytotoxicity in either parent HCT-116 cells or ABCG2-expressing HCT-116/BCRP cells (Table 2). Therefore, PFC is unlikely to be toxic. Moreover, PFC did not alter the expression of *ABCG2* mRNA in HCT116/BCRP cells after exposure to 10 μM PFC for 3 d (Figure 3B). Thus, these data demonstrate that PFC targets the ABCG2 transporter at the functional level in cancer cells.

An ATPase assay was carried out to determine whether PFC interacts at the substrate-binding site, as ABCG2 utilizes the energy of ATP-binding and hydrolysis to transport substrates out of the cell. The transport substrates often stimulate ABCG2-mediated ATP hydrolysis [44,45]. Our data suggest that PFC may directly interact with the substrate-binding site of ABCG2 (Figure 3A) and that this compound itself may be an ABCG2 substrate. However, we did not determine if PFC was being transported. To confirm whether PFC is a substrate, a radiolabeled or fluorescence-based transport assay will be required in future studies.

The mouse experiment in this study indicated that PFC reversed ABCG2-mediated MDR (Figure 4A). In addition, no significant adverse effects (neither mortality nor significant weight loss) were observed, indicating that PFC and its combination treatment (SN-38) did not enhance the toxicity. Therefore, PFC shows promise as a potent, non-toxic, and effective chemosensitizer in cancer chemotherapy. However, the metabolic pathways and bioavailability of PFC remain unknown. As such, further investigations will provide information regarding its most appropriate route of administration and dosage.

## 4. Materials and Methods

### 4.1. Screening of Chemical Compounds

The Tohoku University Graduate School of Pharmaceutical Science has a chemical library consisting of 5812 chemical compounds, such as alkaloids, flavones, polyphenols, heterocyclic compounds, and biologically active natural products [33]. Pre-prepared 384-well plates with compounds dissolved in DMSO to a final concentration of 2 mM, with 2 μL aliquoted per well, were provided by the Tohoku University Graduate School of Pharmaceutical Science (Sendai, Japan). Those compounds were diluted and used for the high-throughput screening (with final concentration 10 μM), as described in Section 4.4.

### 4.2. Chemicals

Dulbecco’s Modified Eagle’s Medium (DMEM), Iscove’s Modified Dulbecco’s Medium (IMDM), fetal bovine serum (FBS), PBS without Ca^2+^ and Mg^2+^ (PBS (−)), PBS containing Ca^2+^ and Mg^2+^ (PBS (+)), penicillin-streptomycin, and pheophorbide a (PhA), were purchased from Sigma-Aldrich (St. Louis, MO, USA). Ko143 was purchased from Tocris Bioscience (Bristol, UK). SN-38 was purchased from Wako Pure Chemical Industries (Tokyo, Japan). Irinotecan was purchased from Yakult Honsha (Tokyo, Japan). CellTiter 96 AQueous One Solution Reagent was purchased from Promega (Madison, WI, USA).

### 4.3. Cell Lines

HCT-116 human colon cancer cells and HCT-116/BCRP cells, which overexpress ABCG2 by transduction of ABCG2 into HCT-116 cells using a retrovirus [46,47], were cultured in DMEM supplemented with 10% FBS and 1% penicillin-streptomycin at 37 °C in 5% CO_2_. The HCT-116/BCRP cells were provided by Dr. Yoshikazu Sugimoto (Keio University, Tokyo, Japan).

### 4.4. High-Throughput Screening

Screens were performed as described previously [29,30]. To maintain identical concentration of compounds, 78 µL of PBS (−) was added to each well of 384-well compound plates, before screening. HCT-116/BCRP cells were transferred at 1 × 10^4^ cells/well in 30 µL into black-walled, clear-bottomed 384-well plates using a Multidrop Combi (Thermo Fisher Scientific, Waltham, MA, USA). After four hours of incubation, PhA (10 µL) and screening compounds (10 µL) were added and incubated for 18 h. In addition to screening compounds, Ko143 (10 µL) and PBS (−) (10 µL, Thermo Fisher Scientific, Waltham, MA, USA) were used as controls. To dispense the chemicals, a Biomek NXP (Beckman Coulter, Indianapolis, IN, USA) was used. The final concentrations of PhA, screening compounds, and Ko143 were 2, 10, and 1 µmol/L, respectively. After removal of the medium and washing with PBS (+), fluorescence intensity was read on a SpectraMax M2e (Molecular Devices, Sunnyvale, CA, USA) in bottom-read mode with 395 nm excitation and 670 nm emission.

### 4.5. Flow Cytometry

To assess the ability of the candidate compounds to inhibit ABCG2-mediated transport, flow cytometry was performed as described previously [19]. HCT-116/BCRP cells were incubated with each candidate compound (or Ko143) and PhA for 30 min at 37 °C in IMDM supplemented with 10% FBS, then washed and incubated for 1 h with each candidate compound in the absence of PhA. After incubation, cells were washed and resuspended in phosphate-buffered saline (PBS). The final concentrations of PhA and Ko143 were 10 µM and 1 µM, respectively. The final concentrations of the candidate compounds were 1, 5, and 10 µM. Fluorescence intensity was immediately measured using BD FACS Verse and BD Suite software (Franklin Lakes, NJ, USA).

### 4.6. Cytotoxic Assay

The effect of the candidate compounds on anticancer drug cytotoxicity in HCT-116 and HCT-116/BCRP cells was determined using the CellTiter 96 AQueous One Solution cell proliferation assay (MTS assay). In brief, 1.0 × 10^4^ cells were seeded in 96-well plates and cultured for 24 h. Subsequently, multiple concentrations of SN-38 were added in 4 ways: with 5 or 10 µM (final concentration) of candidate compounds or with 1 μM of Ko143 and without, then incubated for 72 h. After incubation, 20 µL of MTS solution was added to each well and incubated for 1 h. Absorbance was then measured using a Multiskan FC (Thermo Fisher Scientific). Viability was calculated as: [(absorbance in candidates with anticancer drug) − (absorbance in blank)]/[(absorbance in candidates only) − (absorbance in blank)]. IC_50_ values were calculated using JMP Pro 14.0 (Cary, NC, USA).

### 4.7. Preparation of the Phenylfurocoumarin Derivative

The compound (R)-9-(3,4-dimethoxyphenyl)-4-((3,3-dimethyloxiran-2-yl)methoxy)-7H-furo [3,2-g]chromen-7-one (PFC), was obtained from the diversity-enhanced extracts from the roots of *Angelica dahurica* [48,49]. One kilogram of *A. dahurica* roots, purchased from UCHIDA WAKANYAKU Ltd. (Tokyo, Japan), was extracted twice with methanol (6.5 L) at room temperature. This initial extract was further extracted using ethyl acetate and a saturated sodium bicarbonate solution to yield an ethyl acetate solute (45 g). This solute (17 g) was chromatographed over silica gel and the column was eluted with hexane-ethyl acetate (1:2) mixtures to obtain the furocoumarin-rich fraction (5.63 g). This fraction (1.02 g) was then dissolved in acetonitrile (20 mL), then acetic acid (1 mL) and *N*-bromosuccinimide (787 mg) were added at 0 °C. After stirring for 3 h at 0 °C, the reaction mixture was poured into water and extracted three times with ethyl acetate. The combined organic layer was washed with water and brine, dried over sodium sulfate, and concentrated in vacuo to obtain a crude brominated solute (1.68 g). The crude solute (296 mg) was dissolved in tetrahydrofuran (6 mL) in an argon atmosphere. Then, 3,4-dimethoxyphenylboronic acid (174 mg), [1,1-bis(diphenylphosphino)ferrocene]dichloropalladium(II)–dichloromethane adduct (50 mg) and potassium carbonate (320 mg) were added to the solution. After being refluxed for 12 h, the reaction mixture was cooled to room temperature and filtered through a Celite^®^ pad, then eluted with ethyl acetate. The filtrate was concentrated in vacuo to obtain diversity-enhanced extracts containing PFC.

Thereafter, the diversity-enhanced extracts (500 mg) were chromatographed over silica gel and the column was eluted with hexane-ethyl acetate (1:1) mixtures to obtain crude PFC (34 mg). The crude PFC was subjected to separation using HPLC (column, YMC-GPC T-2000 (φ 20 mm × 600 mm, YMC, Kyoto, Japan); solvent, ethyl acetate) to yield purified PFC (22 mg). The PFC synthesis scheme is described in Figure 5.

### 4.8. ATPase Assay

ABCG2-mediated ATP hydrolytic activity was evaluated using ABCG2-expressing membrane vesicles, as previously described [44]. After the addition of varying concentrations of PFC, ABCG2-expressing membrane vesicles (6 µg/tube) were incubated in ATPase buffer with and without sodium orthovanadate (Vi). The ATPase reaction was initiated by the addition of 5 mM ATP and the mixture was incubated for 20 min at 37 °C. Sodium dodecyl sulfate solution (5%, 0.1 mL/tube) was added to stop the reaction and the amount of inorganic phosphate released was quantified by the colorimetric method, as described previously [45].

### 4.9. Reverse Transcription-Quantitative Polymerase Chain Reaction (RT-qPCR)

HCT-116/BCRP cells were cultured in 10 µM of PFC for 3 d. Then, RNA was isolated from HCT-116 cells, HCT-116/BCRP cells, and HCT-116/BCRP cells with PFC, using the RNeasy Mini Kit (Qiagen, Hilden, Germany). RT-qPCR was performed as previously described [50]. The primers used for *ABCG2* were 5′-GGTCAGAGTGTGGTTTCTGTAGCA-3′ (forward) and 5′-GTGAGAGATCGATGCCCTGCTTTA-3′ (reverse), and the primers for *GAPDH* were 5′-GCACCGTCAAGGCTGAGAAC-3′ (forward) and 5′-TGGTGAAGACGCCAGTGGA-3′ (reverse). Data were analyzed using the 2^−ΔΔCt^ method and normalized to the amount of *GAPDH* mRNA.

### 4.10. In Vivo Antitumor Activity

HCT-116/BCRP cells (4 × 10^6^ cells) in 0.2 mL PBS (−) and 0.1 mL Matrigel Matrix (Corning, NY, USA) were mixed and subcutaneously injected into the right and left shoulders of 8–10-week-old BALB/c nu/nu mice (CLEA Japan, Inc., Tokyo, Japan), to study the chemosensitizing properties of PFC in vivo. Mice were randomized into 4 groups (*n* = 5) after their tumors attained diameters of 0.5–1.0 cm: (a) control (saline), (b) PFC (10 mg/kg), (c) irinotecan (30 mg/kg), and (d) irinotecan (30 mg/kg) + PFC (10 mg/kg). Irinotecan was diluted with saline and injected intraperitoneally at 30 mg/kg (final volume 400 mL). PFC was dissolved in DMSO and diluted in saline to 5% *w/v* and injected intraperitoneally at 10 mg/kg (final volume 400 mL). Injections were repeated seven times on every third day (0, 3, 6, 9, 12, 15, and 18). Body weights and tumor volumes were measured every 3 days. Tumor volumes were calculated using the formula: estimated tumor volume (mm^3^) = 0.5 × (long diameter) × (short diameter)^2^ [51]. This experiment was approved by the Animal Care and Use Committee of Tohoku University Graduate School of Medicine (2019MdA-164). All mice were handled according to the Guidelines for the Care and Use of Laboratory Animals of Tohoku University.

### 4.11. Statistical Analysis

For screening and validation assays, at least three independent experiments were conducted with the identified candidate compounds. Data are presented as means ± SE. Student’s *t*-test was performed to assess the differences between two means. Results were considered statistically significant at *p* < 0.05. Statistical analyses were conducted using JMP Pro 14.0 software.

## 5. Conclusions

This study suggests that PFC inhibits ABCG2-mediated drug-transport function by directly interacting with its substrate-binding sites. It was also revealed that PFC inhibited tumor growth in a murine xenograft model with no adverse effects. Therefore, PFC has promise as a therapeutic to overcome ABCG2-mediated MDR to improve the efficiency of cancer chemotherapy.

## Figures and Tables

**Figure 1 ijms-22-12502-f001:**
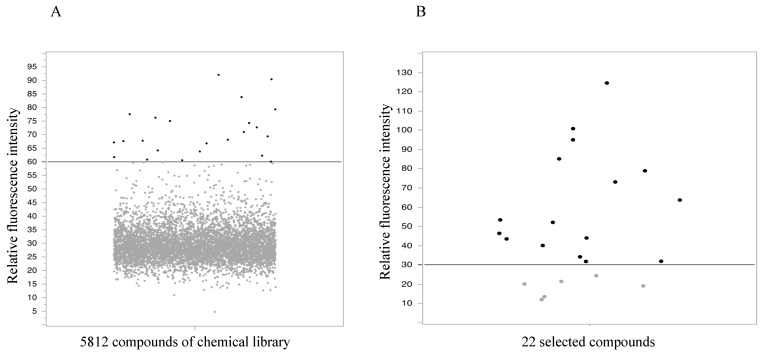
High-throughput screening and validation assay identifying 16 candidate ABCG2 inhibitors. (**A**) Dot plot of PhA efflux data from high-throughput screening. Of 5812 compounds, 23 exhibited higher fluorescence intensities than Ko143. (**B**) Dot plot of PhA efflux data for validation by flow cytometry. Of 22 compounds (there was an overlap for one of the compounds, the number of candidates was reduced to 22), 16 showed higher fluorescence intensities than Ko143.

**Figure 2 ijms-22-12502-f002:**
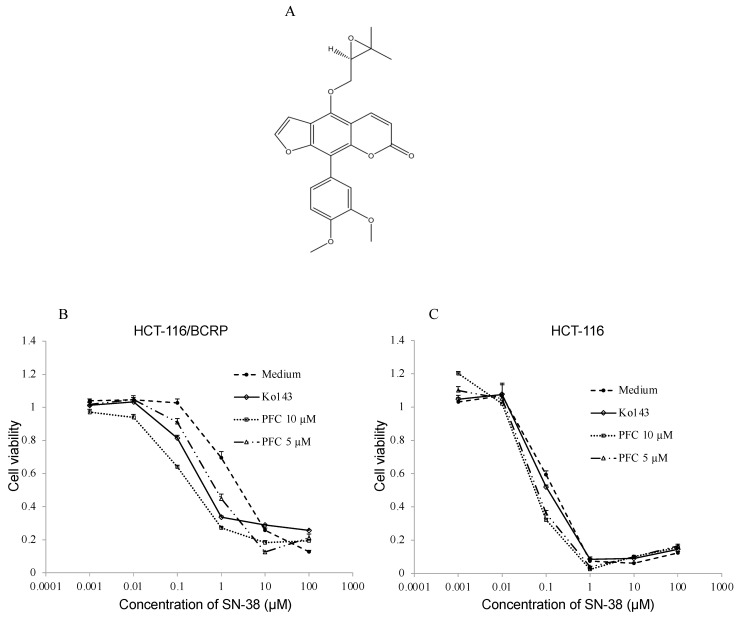
(**A**) Structure of a phenylfurocoumarin derivative (PFC). (**B**,**C**) Effect of PFC on SN-38 cytotoxicity in HCT-116/BCRP (**B**) and HCT-116 (**C**) cell lines. Cells were incubated in the presence of multiple concentrations of SN-38 with 5 or 10 µM of PFC, or 1 µM of Ko143, or medium only (control). Viability was determined by triplicate MTS assays. These figures show the result of one representative experiment of three independent experiments. Points, mean (*n* = 5); bars, SD.

**Figure 3 ijms-22-12502-f003:**
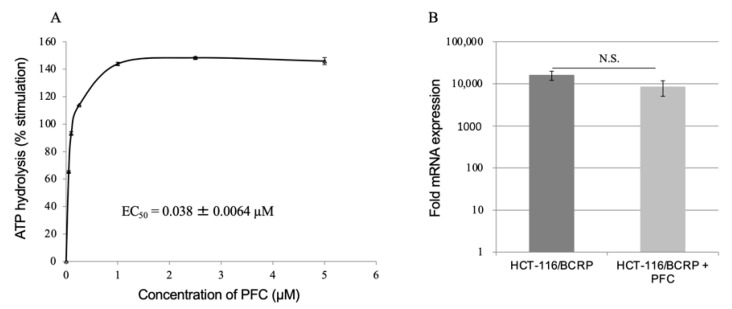
Effect of the phenylfurocoumarin derivative PFC on ATPase activity of ABCG2 and mRNA expression of ABCG2. (**A**) Representative data from four independent experiments. Each point represents a mean (*n* = 4). The concentration of PFC required for 50% stimulation (EC50) is shown as mean ± SE. (**B**) *ABCG2* mRNA expression in HCT-116/BCRP and HCT-116/BCRP with PFC was measured by qPCR, relative to that of HCT-116. Data are shown as mean ± standard error (SE) (*n* = 3). N.S. not significant.

**Figure 4 ijms-22-12502-f004:**
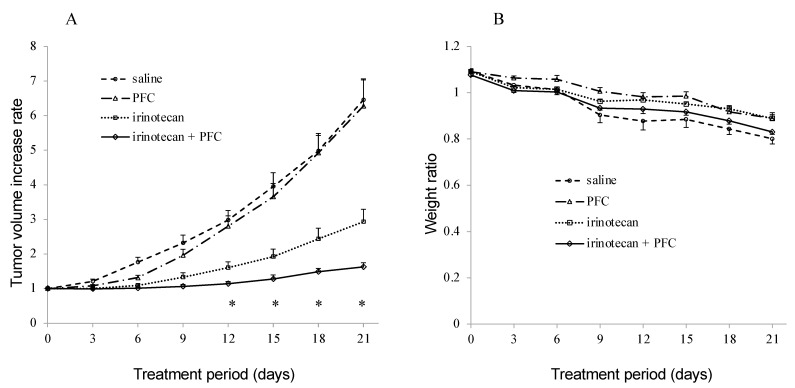
The phenylfurocoumarin derivative PFC reverses resistance to irinotecan in HCT-116/BCRP cell xenografts in nude mice. Tumor volume increase rate (**A**) and body weight rate (**B**) were measured every 3 days. Each point represents a mean (*n* = 5). Bars show standard error (SE). *p* values were determined with the two-tailed Student’s *t*-test (* *p* < 0.05: irinotecan versus irinotecan + PFC).

**Figure 5 ijms-22-12502-f005:**
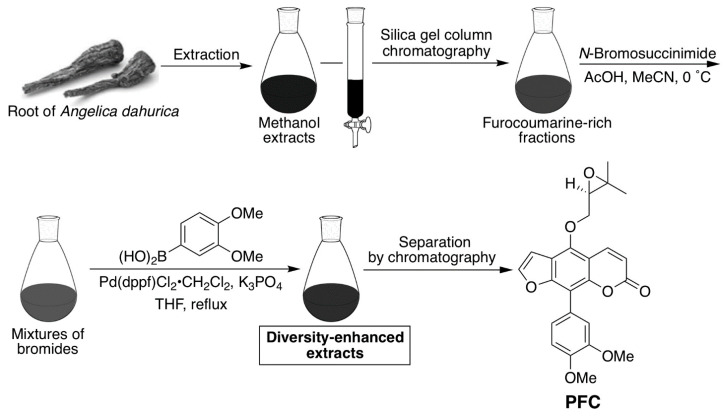
PFC synthesis scheme.

**Table 1 ijms-22-12502-t001:** The phenylfurocoumarin derivative PFC reverses ABCG2-mediated drug resistance to SN-38.

	IC_50_ ± SE (µM) ^a^	Fold Reversal ^b^	IC_50_ ± SE (µM) ^a^	Fold Reversal ^b^
	HCT-116		HCT-116/BCRP	
Medium	0.11 ± 0.0037	1.0	2.59 ± 0.24	1.0
Ko143 1 µM	0.11 ± 0.0036	1.0	0.26 ± 0.027	10.1
PFC 5 µM	0.077 ± 0.022	1.4	0.36 ± 0.11	7.3
PFC 10 µM	0.057 ± 0.013	1.9	0.19 ± 0.018	13.8

^a^ Values are mean ± SE. ^b^ The fold reversal of MDR was calculated by dividing the IC_50_ for cells with SN-38 (medium) in the absence of the reversing agent, by that obtained in the presence of that agent.

**Table 2 ijms-22-12502-t002:** Cytotoxicity of Ko143 and PFC.

	IC_50_ ± SE (µM) ^a^	
	HCT-116	HCT-116/BCRP
Ko143 1 µM	77.97 ± 2.33	115.79 ± 4.85
PFC 10 µM	42.70 ± 2.42	43.07 ± 3.18

^a^ Values are mean ± SE of three independent experiments performed in triplicate.

## Data Availability

The datasets used and/or analyzed during the current study are available from the corresponding author upon reasonable request.

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
