# Peer review of "A Phenylfurocoumarin Derivative Reverses ABCG2-Mediated Multidrug Resistance In Vitro and In Vivo"

_ijms, 2021, doi:10.3390/ijms222212502_

Round 1
Reviewer 1 Report
The aim of this study was to select efficient novel ABCG2 inhibitors and reveal their effect on ABCG2-mediated chemosensitivity in vitro in colon cancer cells. The authors found phenylfurocoumarin derivative (PFC) as effective inhibitor of ABCG2 in vitro and also in vivo in HCT-116/BCRP cell xenograft mouse model. This compounds seems to be potentially clinically interesting to overcome ABCG2-mediated MDR in colon cancer patients. The manuscript is very well and clearly written with detailed methodological description. It is capable of being published after the minor revision process.
Minor points:
- Methods; paragraph 2.9. Please include information how exactly was qRT-PCR method analyzed? Using delta Ct method?
- Methods; paragraph 2.10. Authors should include information describing the exact composition of solutions used for in vivo application. How exactly were dissolved irrinotecan and PFC for in vivo i.p. application? What the final concentration of DMSO was used?
- Supplementary material: Title of the table: Effect of a phenylfurocoumarin PFC "derivatives" on reversing ....
Author Response
We thank the reviewers for their appreciation of our work, and for the helpful comments on the manuscript. We accordingly changed the manuscript, figures and tables to address the reviewer’s Comments and the formatting requirements of the Journal. The modified parts are highlighted in yellow in the manuscript.
Response to reviewer’s comments.
- Comments and Suggestions for Authors from Reviewer 1
- Methods; paragraph 2.9. Please include information how exactly was qRT-PCR method analyzed? Using delta Ct method?
Data were analyzed using the 2−ΔΔCt method and normalized to the amount of GAPDH mRNA. This sentence has been added to the section of “4.9. Reverse Transcription-Quantitative Polymerase Chain Reaction” in Materials and Methods.
- Methods; paragraph 2.10. Authors should include information describing the exact composition of solutions used for in vivo application. How exactly were dissolved irrinotecan and PFC for in vivo i.p. application? What the final concentration of DMSO was used?
Irinotecan was diluted with saline and injected intraperitoneally at 30 mg/kg (final volume 400 ml). PFC was dissolved in DMSO and diluted in saline to 5% w/v and injected intraperitoneally at 10 mg/kg (final volume 400 ml). This information is added to the section of “4.10. In Vivo Antitumor Activity” in Materials and Methods.
- Supplementary material: Title of the table: Effect of a phenylfurocoumarin PFC "derivatives" on reversing ....
Thank you for the correction. This supplementary table shows the results of cytotoxic assays with the 16 candidate compounds on SN38 cytotoxicity in HCT-116/BCRP cells. These 16 compounds were selected after the screening process of the high-throughput screening and the flow cytometry assays. Therefore, the title of the supplementary table (Table S1) was corrected as follows; “The effect of candidate compounds on the reversal of ABCB2-mediated resistance to SN38 in HCT-116/BCRP cells”.
Reviewer 2 Report
The manuscript of dr. Ohnuma and colleagues concerns about the identification of a novel phenylfurocoumarin derivative able to reverse the ABCG2-mediated multidrug resistance in vitro and in vivo.
The manuscript is interesting and covers an important field in anticancer research: multidrug resistance.
The manuscript deserves to be published but in its actual form, it needs some substantial revisions before continuing the publishing route.
Major:
- In the abstract and in the Introduction, such as in the conclusion it would be appropriate to insert the IUPAC name of the PFC derivative
- line 50: the authors refer to ABCG2 substrates, they probably would say inhibitors?
- line 78: "All compounds were dissolved in DMSO to a final concentration of 2 mM" but the authors report different concentrations in the following paragraphs, 10 uM. 2 mM could be a toxic [ ]
- About the concentrations used, why did the authors used different [ ] " The final concentrations 104 of PhA, screening compounds, and Ko143 were 2, 10, and 1 μmol/L, respectively" If Ko143 is the reference compound they should have used the same concentration for the HTS compounds
- Preparation of PFC: it would be appropriate to introduce a synthetic scheme of the PFC derivative and the final structure at this level
- It is not clear the use of 2 different concentrations in 5 and 10 uM in the cytotoxic assay. Usually, the IC50 is used for the other assay. Figure 1 and Table 1 are not clear.
- The major issue concerns the discussion section: the authors cite a compound known as oxypeucedanin, but the role of this compound is not reported in the results section
Author Response
We thank the reviewers for their appreciation of our work, and for the helpful comments on the manuscript. We accordingly changed the manuscript, figures and tables to address the reviewer’s Comments and the formatting requirements of the Journal. The modified parts are highlighted in yellow in the manuscript.
Comments and Suggestions for Authors from Reviewer 2
- In the abstract and in the Introduction, such as in the conclusion it would be appropriate to insert the IUPAC name of the PFC derivative
As suggested by the reviewer, the IUPAC name of PFC ((R)-9-(3,4-dimethoxyphenyl)-4-((3,3-dimethyloxiran-2-yl)methoxy)-7H-furo[3,2-g]chromen-7-one) has been added to the Abstract.
- line 50: the authors refer to ABCG2 substrates, they probably would say inhibitors?
Thank you for pointing out the error. We corrected the word from “substrates” to “inhibitors” in the revised version.
- line 78: "All compounds were dissolved in DMSO to a final concentration of 2 mM" but the authors report different concentrations in the following paragraphs, 10 uM. 2 mM could be a toxic.
As suggested additional details are provided. Pre-prepared 384-well plates with compounds dissolved in DMSO to a final concentration of 2 mM, with 2 μL aliquoted per well were provided by the Tohoku University Graduate School of Pharmaceutical Science (Sendai, Japan). The compounds were diluted and used for the high-throughput screening (with final concentration 10 mM). We added this information to the section of “4.1. Screening of Chemical Compounds” in Materials and Methods.
- About the concentrations used, why did the authors used different [ ] " The final concentrations of PhA, screening compounds, and Ko143 were 2, 10, and 1 μmol/L, respectively" If Ko143 is the reference compound they should have used the same concentration for the HTS compounds
Thank you for the comment. Based on our previous study (Sugisawa N, et al. Mol Pharm. 2018), the concentration of the compounds for high-throughput screening (HTS) was determined to be 10 µM. Ko143 is a known inhibitor of ABCG2 and has been confirmed to be sufficiently effective at 1 µM to inhibit ABCG2-mediated transport function. Therefore, 1 µM Ko143 was used as a positive control. PhA is a known fluorescent ABCG2 substrate, which was used to measure the efflux of the substrate by ABCG2 expressing cells. Although 1 µM of PhA should be sufficient for the HTS assay (Henrich CJ, et al. Mol Cancer Ther, 2007), 1 µM PhA did not show enough fluorescence intensity in our optimization experiments. Therefore, 2 µM PhA was used in this study.
- Preparation of PFC: it would be appropriate to introduce a synthetic scheme of the PFC derivative and the final structure at this level
As the reviewer suggested, we made a revised scheme of the PFC as shown in Figure 5.
- It is not clear the use of 2 different concentrations in 5 and 10 uM in the cytotoxic assay. Usually, the IC50 is used for the other assay. Figure 1 and Table 1 are not clear.
Thank you for the comments. To confirm the dose-dependent inhibitory effect of PFC on the reversal of ABCB2-mediated resistance to SN38 in HCT-116/BCRP cells, cytotoxic assays were carried out by using two different concentrations of PFC in 5 and 10 µM. As shown in Figure 2B, 10 µM PFC significantly decreased the IC50 of SN-38 in HCT-116/BCRP cells compared to 5 µM PFC or 1µM Ko143. Figure caption of Figure 1 and Table 1 were also modified.
- The major issue concerns the discussion section: the authors cite a compound known as oxypeucedanin, but the role of this compound is not reported in the results section
Thank you for the critical comment. As mentioned in the Discussion section, oxypeucedanin slightly enhanced the SN-38-induced cytotoxicity; however, its effect was lower than that of PFC. Therefore, we did not show the experimental data of oxypeucedanin in the Result section. As per the reviewer’s suggestion, we added the results of the cytotoxic assay with oxypeucedanin on reversing ABCG2-mediated drug resistance to SN38 as a supplementary table (Table S2). This information has been added in the Discussion Section.
Round 2
Reviewer 2 Report
No further comments